# Novel Glycosylation by Amylosucrase to Produce Glycoside Anomers

**DOI:** 10.3390/biology11060822

**Published:** 2022-05-27

**Authors:** Jiumn-Yih Wu, Hsiou-Yu Ding, Shun-Yuan Luo, Tzi-Yuan Wang, Yu-Li Tsai, Te-Sheng Chang

**Affiliations:** 1Department of Food Science, National Quemoy University, Kinmen County 892, Taiwan; wujy@nqu.edu.tw; 2Department of Cosmetic Science, Chia Nan University of Pharmacy and Science, No. 60 Erh-Jen Rd., Sec. 1, Jen-Te District, Tainan 71710, Taiwan; ding8896@gmail.com; 3Department of Chemistry, National Chung Hsing University, Taichung 40227, Taiwan; syluo@dragon.nchu.edu.tw; 4Biodiversity Research Center, Academia Sinica, Taipei 11529, Taiwan; tziyuan@gmail.com; 5Department of Biological Sciences and Technology, National University of Tainan, Tainan 70005, Taiwan; aa0920281529@gmail.com

**Keywords:** amylosucrase, *Deinococcus geothermalis*, *Ganoderma lucidum*, glycosyl hydrolase, saponin, triterpenoid

## Abstract

**Simple Summary:**

All livings are composed of cells, which contain lipid, proteins, nuclei acids, and saccharides. Saccharides include polysaccharides, oligo saccharides, disaccharides, which are linked by monosaccharides. Monosaccharides such as glucose exist in two forms, named *α* and *β* anomer, in solution. In addition, monosaccharides could be linked with lipid, proteins, nuclei acids or other saccharide to form glycosides through glycosylation. In nature, glycosylation is catalyzed by enzymes. Until now, all enzymes catalyzed glycosylation to form glycosides with either *α* or *β* form but not both. This study found an enzyme, amylosucrase from *Deinococcus geothermalis* (*Dg*AS), could catalyze glycosylation of a kind of lipid named ganoderic acids triterpenoids from a medicinal fungus *Ganoderma lucidum* to form both *α* and *β* anomer of glycosides. This is the first report that enzymes could catalyze such glycosylation and a possible reaction mechanism was proposed.

**Abstract:**

Glycosylation occurring at either lipids, proteins, or sugars plays important roles in many biological systems. In nature, enzymatic glycosylation is the formation of a glycosidic bond between the anomeric carbon of the donor sugar and the functional group of the sugar acceptor. This study found novel glycoside anomers without an anomeric carbon linkage of the sugar donor. A glycoside hydrolase (GH) enzyme, amylosucrase from *Deinococcus geothermalis* (*Dg*AS), was evaluated to glycosylate ganoderic acid F (GAF), a lanostane triterpenoid from medicinal fungus *Ganoderma lucidum*, at different pH levels. The results showed that GAF was glycosylated by *Dg*AS at acidic conditions pH 5 and pH 6, whereas the activity dramatically decreased to be undetectable at pH 7 or pH 8. The biotransformation product was purified by preparative high-performance liquid chromatography and identified as unusual *α*-glucosyl-(2→26)-GAF and *β*-glucosyl-(2→26)-GAF anomers by mass and nucleic magnetic resonance (NMR) spectroscopy. We further used *Dg*AS to catalyze another six triterpenoids. Under the acidic conditions, two of six compounds, ganoderic acid A (GAA) and ganoderic acid G (GAG), could be converted to *α*–glucosyl-(2→26)-GAA and *β*–glucosyl-(2→26)-GAA anomers and *α*-glucosyl-(2→26)-GAG and *β*-glucosyl-(2→26)-GAG anomers, respectively. The glycosylation of triterpenoid aglycones was first confirmed to be converted via a GH enzyme, *Dg*AS. The novel enzymatic glycosylation-formed glycoside anomers opens a new bioreaction in the pharmaceutical industry and in the biotechnology sector.

## 1. Introduction

Glycosylation is the reaction in which a carbohydrate (i.e., a glycosyl donor) attaches to a functional group of another molecule (a glycosyl acceptor) and forms a glycoconjugate or glycoside [1]. The glycosyl acceptor could be a lipid, a protein, or another sugar, and the produced glycoside was found to be linked by an *O*- (an *O*-glycoside), *N*- (a glycosylamine), *S*- (a thioglycoside), or *C*- (a *C*-glycoside) glycosidic bond. In nature, the attached sugar group is bonded through the anomeric carbon (the hemiacetal/hemiketal carbon) of the sugar donor to the functional group of the glycosyl acceptor via a glycosidic bond. According to the anomer type of the attached sugar, the typical enzymatic glycosides could be either an *α*-linkage or *β*-linkage anomer. Different anomers have different physical, chemical, and biological properties. Therefore, anomer glycosylation with different enzymes could be explored in the future.

In general, the glycosylation of natural products is catalyzed by glycosylatransferase (GT), which transfers the sugar moiety from activated sugar donors (sugar nucleotides), such as uridine diphosphate-glucose (UDP-G), to the sugar acceptor molecules [2]. According to a carbohydrate activating enzyme (CAZy) database, there are 117 subfamilies of GTs containing over 850,000 GTs that have been discovered so far [3]. The GT enzymes use the sugar nucleotides to synthesize complex carbohydrates and crucial intermediates in carbohydrate metabolism [4]. However, all of the sugar nucleotides must be linked with anomeric carbon.

Natural triterpenoids, which include more than 14,000 identified triterpenoids, are one of the largest subclasses of specialized metabolites, and usually form tetra- or pentacyclic structures. To date, more than 100 cyclical triterpene scaffolds have been identified from plants and fungus [5]. Triterpenoids exhibit various bioactivities and thus play important roles in developing new drugs. In plants, triterpenoids usually exist in the glycosidic forms, called saponins, with linkages of sugar moieties. Some saponins possess unique bioactivities while their aglycone forms do not. For example, ginseng saponin Rg1 has been reported to increase human immune responses; detox oxidized low-density-lipoproteins relax the endothelium-dependent porcine coronary arteries, improve insulin resistance in diabetes, suppress tumor cell growth in cancers and arrest melanoma cell cycle progression in the G1 phase. Animal model studies have also demonstrated that ginseng saponins Rb1, CK and Rg1 can prevent scopolamine-induced memory deficits. In contrast, the ginseng triterpenoid aglycones did not demonstrate any bioactivity [6]. Thus, the glycosylation of triterpenoids has become a derived area in biotechnology.

On the other hand, triterpenoids from microorganisms rarely occur in glycosidic form. For example, a well-known medicinal fungus, *Ganoderma lucidum*, called ‘Lingzhi’ in Chinese, also contains plenty of triterpenoids, of which over 300 have been identified [7,8,9]. Among the *Ganoderma* triterpenoids, ganoderic acid A (GAA) is a major one. All *Ganoderma* triterpenoids that belonged to the lanostane type were found to possess various bioactivities. In spite of the diversity of *Ganoderma* triterpenoids, only a few *Ganoderma* triterpenoid saponins were identified [7,8,9]. In previous studies, we found that several bacterial GTs are able to glycosylate *Ganoderma* triterpenoids to *Ganoderma* triterpenoid glycosides [10,11,12,13]. These discovered glycosides were all *β*-linkage aglycons. *Bs*GT110 is the most specific bacterial GT among them [10]. *Bs*GT110 showed glycosylation activity toward GAA specifically at pH 6 but lost most activity at pH 8. In contrast, *Bs*GT110 showed glycosylation activity toward 8-hydroxydaidzein, specifically at pH 8, but lost most activity at pH 6.

Previous studies indicated that GT needs expansive sugar donors (eq: UDP-G) for glycosylation, whereas scientists later used glycoside hydrolases (GHs) to glycosylate molecules with cheap sugar donors, such as starch, maltodextrin, maltose, and sucrose. In addition to hydrolytic activity, several GHs were found to trans-glycosylate the sugar moiety from sugar donors to acceptor molecules [14]. According to the CAZy database, there are 171 GH families and over one million GHs. Several of these GH enzymes, such as GH family 13 (GH13) and GH family 68 (GH68), have been proven to glycosylate small molecules [15]. The precursor molecules include aliphatic alcohols, polyols, phenolic compounds, hydroxycinnamic acids, hydroxybenzoic acids, xanthonoids, flavonoids (glycosides), stilbenoids (glycosides), diterpenoid saponins (steviol), and triterpenoid saponins [14]. Thus, GHs could be enzymes with high potential to produce bioactive glycosylated molecules. However, there is no report that GH can glycosylate a triterpenoid aglycone, although over one million GHs have been identified.

This study found that an amylosucrase is capable of glycosylating the triterpenoid aglycone (ganoderic acid). Amylosucrase (AS, E.C. 2.4.1.4) is a versatile sucrose-hydrolyzing enzyme that belongs to GH13 [16,17]. Amylosucrase can synthesize *α*-1,4-glucans from sucrose as a sole substrate and can utilize various small molecules as acceptors. An amylosucrase from *Deinococcus geothermalis* (*Dg*AS) is the most famous and has been proven to glycosylate many small molecules, including arbutin, baicalein, catechin, daidzin, hydroquinone, luteolin, rutin, salicin [16,17], and 8-hydroxydaidzein [18]. *Dg*AS performed optimal sucrose hydrolysis activity at pH 8 and optimal transglycosylation activity with sucrose toward various small molecules between pH 7 to pH 8 [16,17]. However, the enzyme has never been reported to perform optimal transglycosylation activity to small molecules at acidic conditions. In contrast, our previous results revealed that GAA could be glycosylated via GTs under acidic conditions [10]. Accordingly, we are interesting in evaluating whether *Dg*AS could glycosylate the triterpenoids under different pH.

## 2. Materials and Methods

### 2.1. Enzymes and Chemicals

GAA, ganoderic acid G (GAG), ganoderic acid F (GAF), and celastrol were purchased from Baoji Herbest Bio-Tech (Xi'an, Shaanxi, China). GAA-15-*O*-*β*-glucoside [11], GAA-26-*O*-*β*-glucoside [10], and antcin K [19] were obtained from our previous studies. Recombinant *Dg*AS was prepared from our previous study, and the specific sucrose hydrolysis activity of the purified recombinant *Dg*AS was determined to be 6.6 U/mg [20].

### 2.2. Biotransformation

Biotransformation was conducted according to a previous study [18] and briefly described as below: 1 mg/mL of triterpenoid was incubated with 25 µg/mL of *Dg*AS, 100 mM or 1500 mM of sucrose, and 50 mM of different buffers: acetate buffer (pH 5), phosphate buffer (pH 6 and pH 7), and Tris buffer (pH 8) at 40 °C for 24 h. After reaction, the mixture was analyzed using high-performance liquid chromatography (HPLC).

### 2.3. HPLC Analysis

HPLC analysis was conducted according to a previous study [18] and briefly described as below: An Agilent^®^ 1100 series HPLC system (Santa Clara, CA, USA) equipped with a gradient pump (Waters 600, Waters, Milford, MA, USA) was used and the stationary phase was a C18 column (Sharpsil H-C18, 5 μm, 4.6 i.d. × 250 mm, Sharpsil, Beijing, China), and the mobile phase was 1% acetic acid in water (A) and methanol (B). The elution condition was a linear gradient from 0 min with 40% B to 20 min with 70% B; isocratic from 20 min to 25 min with 70% B; a linear gradient from 25 min with 70% B to 28 min with 40% B; and isocratic from 28 min to 35 min with 40% B. The flow rate of the mobile phase was 1 mL/min. The sample volume was 10 µL. The detection intensity was set at 254 nm.

### 2.4. Purification and Identification

The purification process of the biotransformation metabolites was conducted according to a previous study [18] and briefly described as below: The biotransformation was scaled-up to 20 mL containing 1 mg/mL of ganoderic acid (GAF, GAA, or GAG), 25 µg/mL of *Dg*AS, 1500 mM of sucrose, and 50 mM of acetate buffer (pH 5). After incubation t 40 °C for 24 h, the biotransformation products were purified by a preparative YoungLin HPLC system (YL9100, YL Instrument, Gyeonggi-do, Korea) [18]. The elution corresponding to the peak of the product in HPLC was collected, concentrated under vacuum, and then lyophilized. Finally, the structure of the purified compound was confirmed using mass and nucleic magnetic resonance (NMR) spectrometry.

## 3. Results and Discussion

### 3.1. Biotransformation of GAF by DgAS 

To make sure that *Dg*AS could glycosylate triterpenoid aglycone, we first used triterpenoid aglycone with simple carboxyl group GAF as the starting point (Figure 1). This *Ganoderma* triterpenoid substrate contains a C-26 carboxyl group, which is the major functional group that exists in all ganoderic acids. GAF was then incubated with *Dg*AS with 1500 mM of sucrose concentrations under different pH values, at 40 °C for 24 h. After incubation, the reaction mixture was analyzed by HPLC. The results indicated that *Dg*AS could glycosylate GAF to compound (**1**) under acidic conditions of pH 6 and pH 5 (Appendix A), and pH 5 was the optimal condition (Figure 2a). However, the activity decreased extremely and became undetectable at pH 7 or pH 8. We performed three negative controls at pH 5: (1) reaction with the heat-denatured (inactivated) enzyme; (2) reaction without substrates (sucrose); and (3) reaction without GAF substrate. There are no targeted products in the three negative controls. The *Dg*AS was first identified in a GH13 enzyme with novel acidic glycosylation of *Ganoderma* triterpenoids; its glycosylatic activities under different pH levels are similar to the enzymatic condition of *Bs*GT110 glycosylation [10].

We further investigated the effects of temperature, sucrose concentration, and reaction time on the acidic glycosylation activity of *Dg*AS toward GAF. The results showed that the optimal reaction conditions were 100 to 1500 mM of sucrose at 40 °C for 24 h (Figure 2). The maximum yield of compound (**1**) was 47%. However, the number of catalytic cycles by the enzyme before its inactivation was not evaluated. It has been reported that high concentrations of sugar donors would improve the glycosylation activity of GH enzymes [21]. However, sucrose concentrations between 100 mM and 1500 mM did not significantly affect the glycosylation activity of *Dg*AS toward GAF.

### 3.2. Purification and Identification of the Biotransformation Product

Compound (**1**) (10.4 mg) was purified from 20 mL of biotransformation of GAF by *Dg*AS by preparative HPLC. The chemical structure of compound (**1**) was identified using mass and NMR spectral analysis. The mass spectrometer showed an [M + H]^−^ ion peak at *m*/*z*: 731.4 in the electrospray ionization mass spectrum (ESI-MS) implying that compound (**1**) contained one glucosyl moiety attached to the GAF (molecular weight of 570) structure (Appendix A). The structure of compound (**1**) was elucidated by NMR in advance. Compound (**1**) showed the presence of a triterpenoid substituent and an *α*-glucose or a *β*-glucose unit, the hydrolyzed moiety of GAF [22,23]. In addition to the signals of the GAF, 14 proton signals (from 4.27 to 6.05 ppm) and 12 carbon signals (from 62.7 to 96.5 ppm) corresponding to two glucose moiety structures were observed. The attachment of the glucose moiety to C-26 of GAF was confirmed by the heteronuclear multiple bond connectivity (HMBC) correlation from the anomeric proton of Glu H-2′*a* (*δ*_H_ 5.50, d, *J* =3.5, 9.8 Hz) to C-26 (δ_C_ 176.2). The observation of coupling constants (*J* = 3.5 Hz) of the anomeric protons confirmed that the glycosidic bonds of compound (**1**) were in the *α*-configuration, demonstrating the structure of compound (**1**) as *α-*glucosyl-(2→26)-GAF. The large coupling constant (7.7, 9.1 Hz) of the anomeric proton H-2′*β* (5.65 ppm) indicated the *β*-configuration. The cross peak of H-2′*β* with C-26 (5.65/175.4 ppm) in the HMBC spectrum demonstrated that the structure of compound (**1**) is *β*-glucosyl-(2→26)-GAF. In addition, two sets of glucose signals, assignable to *α* and *β*, were found in the ^1^H-NMR and ^13^C-NMR spectrum, which could be grouped by distortionless enhancement by polarization transfer (DEPT), heteronuclear single quantum coherence (HSQC), HMBC, correlation spectroscopy (COSY), and nuclear Overhauser effect spectroscopy (NOESY) spectra (shown in Appendix A). The NMR signals were fully identified, as shown in Appendix A. Compound (**1**) thus confirmed the coexistence of *α*-glucosyl-(2→26)-GAF and *β*-glucosyl-(2→26)-GAF; In addition, according to the NMR analysis, the anomeric ratio (*α*:*β*) of the products was 3:2 (60%:40%). Figure 3 summarizes the biotransformation of GAF mediated by *Dg*AS. 

### 3.3. Evaluation of the Glycosylation Activity of DgAS toward Other Triterpenoids

Based on the novel glycosylation activity by *Dg*AS toward the carboxyl group of the tetracyclic triterpenoid GAF, the other either tetracyclic or pentacyclic triterpenoids (antcin K, celastrol and 4 ganoderic acids in Figure 4) with carboxyl group were selected to evaluate the detailed acidic glycosylation activity of *Dg*AS. Among the six triterpenoids, GAA and GAG are the tetracyclic lanostane type of *Ganoderma* triterpenoids with the same C-26 carboxyl group as that of GAF. GAA-15-*O*-*β*-glucoside and GAA-26-*O*-*β*-glucoside were previous biosynthetic glycosides from GAA by the bacterial GTs [10,11]. Antcin K also contains a C-26 carboxyl group. Antcin K is the most abundant tetracyclic ergostane type of triterpenoid isolated from the fruiting bodies of *Antrodia cinnamomea*, which is a parasitic fungus that only grows on the inner heartwood wall of the aromatic tree *Cinnamomum kanehirai* Hay (Lauraceae) [19]. In addition, celastrol contains a C-29 carboxyl group which is a pentacyclic triterpenoid isolated from the root of *Tripterygium wilfordii* (thunder god vine) [24].

The results showed that *Dg*AS could glycosylate GAA and GAG into compound (**2**) and compound (**3**) but cannot glycosylate GAA-15-*O*-*β*-glucoside, GAA-26-*O*-*β*-glucoside, antcin K, or celastrol (Appendix A and Table 1). The glycosylation condition of *Dg*AS toward GAA and GAG was similar to that toward GAF, for which glycosylation activity was favored at pH 5 and pH 6 but dramatically decreased at pH 7 or pH 8, and the acidic activity was not significantly affected by sucrose concentration from 100 to 1500 mM. Although all the six tested triterpenoids contain a carboxyl group in their structures, it seems that the novel acidic glycosylation activity of *Dg*AS was special toward *Ganoderma* triterpenoids.

### 3.4. Purification and Identification of Biotransformation Products from GAA and GAG 

To identify the chemical structure of compound (**2**) and compound (**3**), the biotransformation was scaled up to 20 mL, and the two compounds were purified by preparative HPLC. Finally, 8.8 mg of compound (**2**) and 9.3 mg of compound (**3**) were obtained. The chemical structures of compound (**2**) and compound (**3**) were identified using mass and NMR spectral analysis.

In the mass analysis of compound (**2**), the mass spectrometer showed an [M + H]^−^ ion peak at *m*/*z*: 677.6 in the ESI-MS corresponding to the molecular formula C_36_H_54_O_12_ (Appendix A). The mass data imply that compound (**2**) contained one glucosyl moiety attached to the GAA (molecular weight of 516) structure.

In the mass analysis of compound (**3**), the mass spectrometer showed an [M + H]^−^ ion peak at *m*/*z*: 693.5 in the electrospray ionization mass spectrum (ESI-MS) corresponding to the molecular formula C_36_H_54_O_13_ (Appendix A). The mass data imply that compound (**3**) contained one glucosyl moiety attached to the GAG (molecular weight of 532) structure.

To identify the structures of compound (**2**) and compound (**3**) in advance, the NMR spectroscopy method was then used. The full assignments of the ^1^H and ^13^C- NMR signals were further aided by DEPT, HSQC, HMBC, COSY, and NOESY spectra, as shown in Appendix A.

Compound (**2**) of the NMR spectra exhibited characteristic *α*-glucosyl and *β*-glucosyl signals. In addition to the signals of the GAA [25], 14 proton signals (from 4.28 to 6.04 ppm) and 12 carbon signals (from 62.7 to 96.5 ppm) corresponding to two glucose moiety structures were observed. The attachment of the glucose moiety to C-26 of GAA was confirmed by the HMBC correlation from the anomeric proton of Glu H-2′*a* (*δ*_H_ 5.48, d, *J* = 3.5, 9.8 Hz) to C-26 (*δ*_C_ 176.2). The observation of coupling constants (*J* = 3.5 Hz) of the anomeric protons confirmed that the glycosidic bonds of compound (**2**) were in the *α*-configuration, demonstrating the structure of compound (**2**) as *α*-glucosyl-(2→26)-GAA. The large coupling constant (8.4 Hz) of the anomeric proton H-2′*β* (5.64 ppm) indicated the *β*-configuration. The cross peak of H-2′*β*with C-26 (5.64/175.5 ppm) in the HMBC spectrum demonstrated that the structure of compound (**2**) is *β*-glucosyl-(2→26)-GAA. In addition, two sets of glucose signals, assignable to *α* and *β*, full assignments of the ^1^H- and ^13^C-NMR signals were further aided by DEPT, HSQC, HMBC, COSY, and NOESY spectra, as shown in Appendix A. Compound (**2**) thus confirmed the coexistence of *α*-glucosyl-(2→26)-GAA and *β*-glucosyl-(2→26)-GAA.

Compound (**3**) showed the presence of a triterpenoid GAG [26], and two sets of glucose signals, assignable to *α* and *β*, were found in the ^1^H-NMR and ^13^C-NMR spectra. In addition to the signals of the GAG, 14 proton signals (from 4.21 to 6.05 ppm) and 12 carbon signals (from 62.7 to 96.5 ppm) corresponding to two glucose moiety structures were observed, which could be grouped by HMBC correlations. An ether linkage between H-2′*α* of glucose and C-26 (5.47/176.3 ppm) of the GAG was proven by the HMBC spectrum. The coupling constant (3.5, 9.8 Hz) of the anomeric proton H-2′*α* (5.47 ppm) indicated that the *α*-configuration of the structure of compound (**3**) is *α*-glucosyl-(2→26)-GAG. The large coupling constant (8.4 Hz) of the anomeric proton H-2′*β* (5.64 ppm) indicated the *β*-configuration. The cross peak of H-2′*β* with C-26 (5.64/176.5 ppm) in the HMBC spectrum demonstrated that the structure of compound (**3**) is *β*-glucosyl-(2→26)-GAG. The NMR spectroscopic data are exhibited in Appendix A. The results also showed the coexistence of *α*-glucosyl-(2→26)-GAG and *β*-glucosyl-(2→26)-GAG (**3**). In addition, according to the NMR analysis, the anomeric ratio (*α*:*β*) of the products was 3:2 (60%:40%).

Figure 5 summarizes the biotransformation of GAA and GAG mediated by *Dg*AS. 

### 3.5. Proposed Reaction Mechanism of DgAS toward Ganoderic Acids

In nature, enzymatic glycosylation is always bonded through the anomeric carbon of the sugar donor (C1 of glucose) to the functional group of the glycosyl acceptor. However, this study discovered a new mechanism by which the *Dg*AS catalyzed glycosylation through the non-anomeric carbon (C2) of the sugar donor linkage to the sugar acceptors (ganoderic acids). The novel linkage of the enzymatic glycosylation might occur through a carbon switch mechanism [27]. Figure 6 illustrates the putative transglycosylation mechanism of *Ganoderma* triterpenoids by the GH13 enzyme (*Dg*AS). The glycosyl-enzyme intermediate approaches by ganoderic acid formed an intermediate **I**, which underwent formation of 1,2-cyclic intermediate **II.** The rearrangement of the 1,2-cyclic intermediate then formed the desired glycosylated product [27]. According to the proposed mechanism, the lack of free carboxyl group in GAA-26-*O*-*β*-glucoside explains why the molecule cannot be glycosylated by *Dg*AS (Table 1). Also, an oxo group (C-23) near the glycosylated carboxyl group (C-26) is needed for the stabilization of the intermediate (Figure 6). Thus, neither antcin K nor celastrol is glycosylated by *Dg*AS due to the lack of the oxo group near their carboxyl group. In addition, the bulky glucosyl group at C-15 of GAA-15-*O*-*β*-glucoside might form steric hindrance in the catalytic reaction by *Dg*AS and block the catalytic activity toward the molecule by *Dg*AS.

## 4. Conclusions

*Dg*AS was the first confirmed GH to be able to glycosylate GAA, GAG, and GAF to form glucosyl-(2→26) ganoderic acid anomers under acidic conditions. This study proposed a new mechanism to form the glycoside anomers of triterpenoids by *Dg*AS. The new mechanism by GHs, such as *Dg*AS, to produce the *Ganoderma* triterpenoid saponin anomers could be applied for new triterpenoid derivatives in pharmaceutical biotechnology. 

## Figures and Tables

**Figure 1 biology-11-00822-f001:**
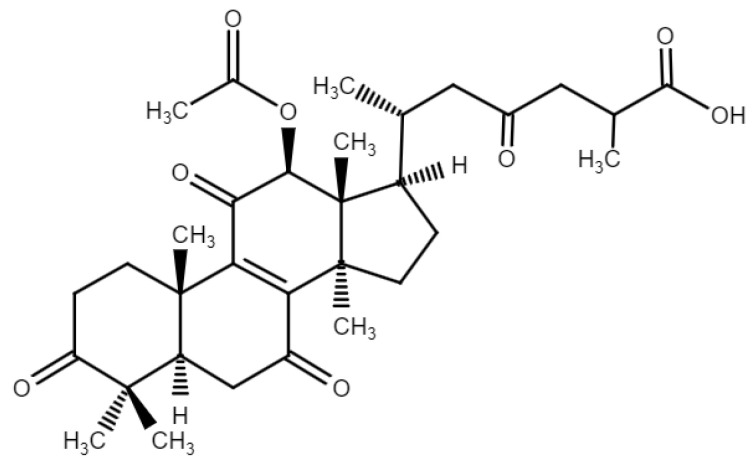
The chemical structure of ganoderic acid F (GAF).

**Figure 2 biology-11-00822-f002:**
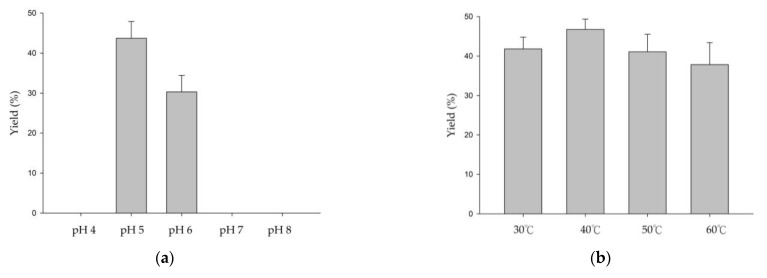
Effects of pH (**a**), temperature (**b**), sucrose concentration (**c**), and reaction time (**d**) on the yield of compound (**1**) from biotransformation of ganoderic acid F (GAF) by *Dg*AS. The standard reaction condition was 1 mg/mL of GAF, 25 µg/mL of *Dg*AS, and 1500 mM of sucrose at 50 mM of acetate buffer (pH 5) and 40 °C for 24 h. To determine suitable reaction conditions, different pH values, temperatures, sucrose concentrations, and reaction times were tested. After incubation, the biotransformation products were analyzed using HPLC. The detailed reaction conditions and the HPLC procedure are described in the Section 2.

**Figure 3 biology-11-00822-f003:**
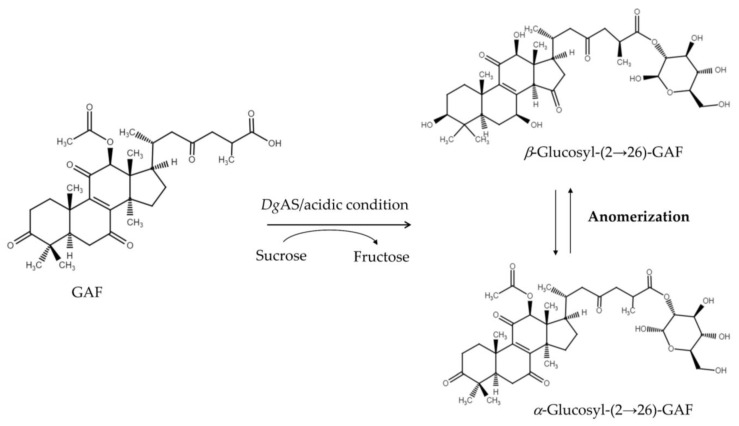
The biotransformation process of GAF to GAF glycoside anomers by *Dg*AS.

**Figure 4 biology-11-00822-f004:**
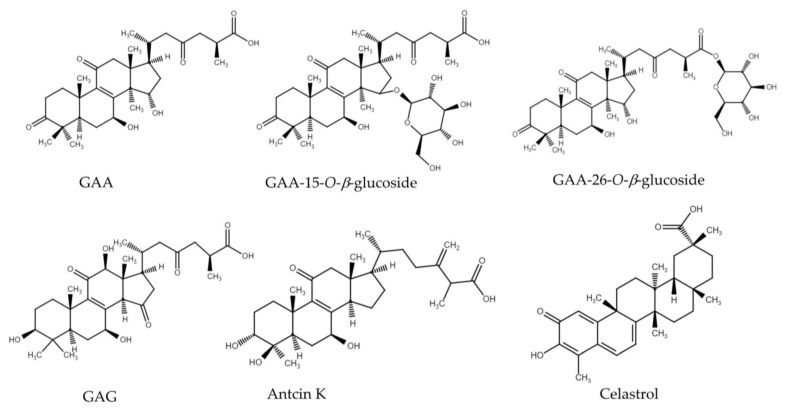
The chemical structures of ganoderic acid A (GAA), ganoderic acid G (GAG), GAA-15-*O*-*β*-glucoside, GAA-26-*O*-*β*-glucoside, antcin K, and celastrol.

**Figure 5 biology-11-00822-f005:**
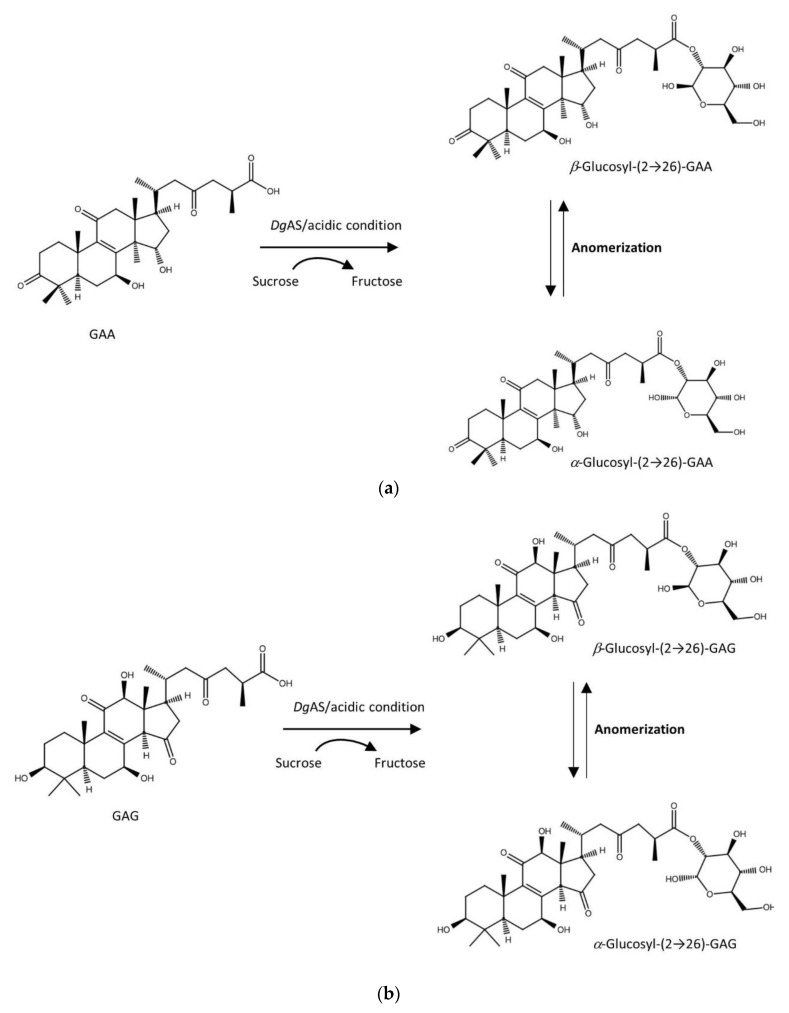
The biotransformation process of GAA (**a**) and GAG (**b**) to the corresponding glycosides by *Dg*AS.

**Figure 6 biology-11-00822-f006:**
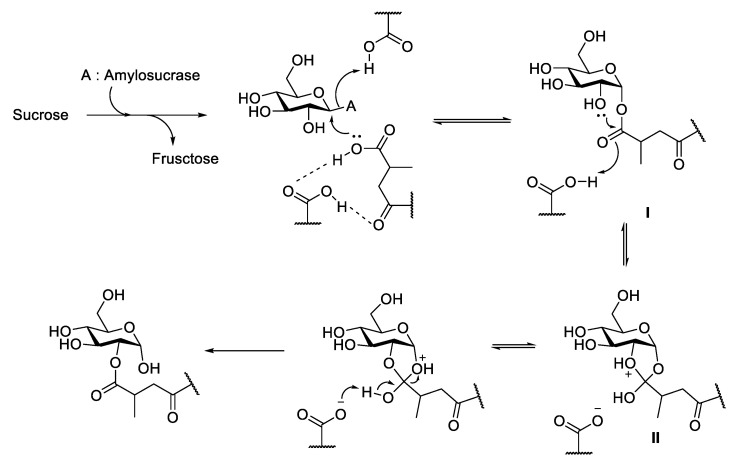
Proposed reaction mechanism of *Dg*AS toward ganoderic acids.

**Table 1 biology-11-00822-t001:** Glycosylation activity ^1^ of *Dg*AS toward tested triterpenoids at different pH and sucrose concentrations.

Triterpenoid	Sucrose (mM)	pH 4	pH 5	pH 6	pH 7	pH 8
GAA	100	N.D. ^2^	53.3 ± 4.7	31.1 ± 2.6	3.1 ± 0.0	N.D.
	1500	N.D.	50.7 ± 2.7	27.7 ± 0.9	3.2 ± 0.1	N.D.
GAG	100	N.D.	52.4 ± 3.6	17.2 ± 1.0	2.1 ± 0.0	N.D.
	1500	N.D.	47.6 ± 1.5	18.0 ± 0.8	0.8 ± 0.0	N.D.
GAA-15-*O*-*β*-glucoside	100	N.D.	N.D.	N.D.	N.D.	N.D.
	1500	N.D.	N.D.	N.D.	N.D.	N.D.
GAA-26-*O*-*β*-glucoside	100	N.D.	N.D.	N.D.	N.D.	N.D.
	1500	N.D.	N.D.	N.D.	N.D.	N.D.
Antcin K	100	N.D.	N.D.	N.D.	N.D.	N.D.
	1500	N.D.	N.D.	N.D.	N.D.	N.D.
Celastrol	100	N.D.	N.D.	N.D.	N.D.	N.D.
	1500	N.D.	N.D.	N.D.	N.D.	N.D.

^1^ Activity was calculated by dividing the amount of the high-performance liquid chromatography (HPLC) area of the produced triterpenoid saponins in each reaction by the HPLC area of the peak of the triterpenoid without enzymes and expressed as a percentage. The mean (*n* = 3) is shown, and the standard deviations are represented by error bars. ^2^ N.D. means not detectable.

## Data Availability

Not applicable.

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
