# Peer review of "Novel Glycosylation by Amylosucrase to Produce Glycoside Anomers"

_biology, 2022, doi:10.3390/biology11060822_

Round 1

Reviewer 1 Report

This is a good piece of work. The authors used GH13 amylosucrase from Deinococcus geothermalis to produce novel glucosyl ganoderic acid compounds by glycosylation. The compounds are very interesting because the C-2 position is modified instead of the C-1 anomeric carbon. This finding is applicable not only to the creation of these compounds, but also to the creation of other compounds. Most of the experimental methods and results are robust in supporting the main conclusions. However, I have some suggestions for improvement.

Adding the following to the Results and Discussion section will help to better determine the validity of the reaction mechanism and provide clues to the substrate specificity of this enzyme.

(1) As a negative control, it is better to react with the inactivated enzyme to show that the modification is catalyzed by the enzyme.

(2) Was there no other known example of glycosylation of R-COOH to sugar instead of R-OH, other than this enzyme?

(3) Does this enzyme not react with compounds with carboxy groups such as acetic acid and propionic acid? The author utilized acetate buffers at pH 4 and 5, but was glucose acetylated at position 2 produced?

(4) What is the optimum pH for the hydrolysis of this enzyme? What is the relationship between the optimum pH for hydrolysis and the optimum pH for glycosylation?

(5) Is it possible to discuss the substrate specificity of this enzyme based on its three-dimensional structure? What are the substrate binding sites? Of the four compounds that did not react, GAA-15-O-beta-glucoside and Antcin K seem to react.

Others:

(1) Lines 44-45, Threfore, producing…by enzymes.: The meaning of the sentence is unclear.

(2) Lines 76, 96, 125, 129, 131, 163: Greek letters are garbled.

(3) Doesn’t pH 8 phosphate buffer inhibit the reaction? Are there any phosphorylated compounds?

(4) Figure 3 (c): Change the unit of concentration of sucrose to mM.

Author Response

Point-to-point responses to reviewers’ comments:

The comments from Reviewer #1:

This is a good piece of work. The authors used GH13 amylosucrase from Deinococcus geothermalis to produce novel glucosyl ganoderic acid compounds by glycosylation. The compounds are very interesting because the C-2 position is modified instead of the C-1 anomeric carbon. This finding is applicable not only to the creation of these compounds, but also to the creation of other compounds. Most of the experimental methods and results are robust in supporting the main conclusions. However, I have some suggestions for improvement.

Major concers:

  1. “As a negative control, it is better to react with the inactivated enzyme to show that the modification is catalyzed by the enzyme.”

Response:

Thank you for the comment. “We performed three negative controls: (1) reaction with the heat-denatured (inactivated) enzyme; (2) reaction without substrates (sucrose); (3) reaction without GAF substrate. There is no targeted products in the three negative controls.”

In order to explain clearer, we added the above description at lines from 168 to 171 at page 4 in the Results and Discussion 3.1 sec. of the revised manuscript.

  1. “Was there no other known example of glycosylation of R-COOH to sugar instead of R-OH, other than this enzyme?”

Response:

There are indeed some studies that the other enzymes could glycosylated R-COOH to sugar. For example, our previous study also found that a bacterial GTs (BsGT110) was able to glycosylate GAA to GAA-26-O-b-glucoside [Ref. 10]. BsGT110 showed glycosylation activity toward GAA specifically at pH 6 but lost most activity at pH 8. The above description has been added at lines from 86 to 90 at page 2 in the Introduction of the revised manuscript.

  1. “Does this enzyme not react with compounds with carboxy groups such as acetic acid and propionic acid? The author utilized acetate buffers at pH 4 and 5, but was glucose acetylated at position 2 produced?”

Response:

Thank you for the comment. The acidic glycosylation of DgAS to produce saponin anomers were observed at pH 5 and pH 6, which used acetic acid and phosphate as buffers, respectively. Because the acidic glycosylation of DgAS could be conducted using different buffer systems, therefore, the possible of the reaction involving the used buffering salts such as acetate is ruled out.

  1. “What is the optimum pH for the hydrolysis of this enzyme? What is the relationship between the optimum pH for hydrolysis and the optimum pH for glycosylation?”

Response:

Thank you for the comment. “DgAS performed optimal sucrose hydrolysis activity at pH 8 and optimal transglycosylation activity with sucrose toward various small molecules between pH 7 to pH 8 [reviewed in 16-17]. However, the enzyme has never reported to perform optimal transglycosylation activity to small molecules at acidic conditions.”

In order to explain clearer, we added the above description at lines from 111 to 114 at page 3 in the Introduction of the revised manuscript.

  1. “Is it possible to discuss the substrate specificity of this enzyme based on its three-dimensional structure? What are the substrate binding sites? Of the four compounds that did not react, GAA-15-O-beta-glucoside and Antcin K seem to react.”

Response:

Thank you very much for the comment. The new mechanism of glycosylation by amylosucrose in the present is indeed worth to study. We are still working on it due to some technical problems.

Minor concers:

  1. “Lines 44-45, Threfore, producing…by enzymes.: The meaning of the sentence is unclear.”

Response:

Thank you for the comment. We rewrote the sentence to “Therefore, anomer glycosylation with different enzymes could be explored in the future.” at lines from 44 to 45 at page 1 in the Introduction of the revised manuscript.

  1. Lines 76, 96, 125, 129, 131, 163: Greek letters are garbled.”

Response:

Thank you for the comments. We corrected all the garbled letters to the correct Greek letters in the revised manuscript.

  1. Doesn’t pH 8 phosphate buffer inhibit the reaction? Are there any phosphorylated compounds?”

Response:

Thank you for the comment. We used Tris buffer to evaluate the glycosylation by DgAS at pH 8 and did not observe any products in the HPLC analysis (Figure S1 and Table 1 in the revised manuscript). Therefore, the possibility of the reaction was inhibited by phosphate at pH 8 was ruled out. Besides, the acidic glycosylation of DgAS to produce saponin anomers were observed at both pH 5 and pH 6, which used acetic acid and phosphate as buffers, respectively, not only using acetic acid or phosphate. Moreover, the acidic glycosylation of ganoderic acids by DgAS to producer significant products were purified and identified by mass and NMR methods. We didn’t see any significant phosphorylated products.

  1. “Figure 3 (c): Change the unit of concentration of sucrose to mM.”

Response:

We changed the unit of concentration of sucrose to mM in the revised figure 2c (the original figure 3c) in the revised manuscript.

Reviewer 2 Report

In this article, Te-Sheng Chang and co-workers report the glycosylation of triterpenoid aglycones via DgAS catalysis. All compound structures were supported by spectroscopic data and the manuscript is easy to follow although some spelling errors were found. Having said the above, I recommend the manuscript for publication in Biology after addressing a number of recommendations:

- It would be desirable to include in the introduction some comparative examples of saponins that display greater biological activity than their aglycones.

- To which structural characteristics in the substrate is attributed that the reaction does not proceed with the remaining triterpenoids? was only the standardized method evaluated or were different conditions also tested?

- Please, detail the criteria for the selection of the 7 aglycones, particularly antcin K and celastrol.

- Were studies conducted to determine the number of catalytic cycles given by the enzyme before its inactivation? Please indicate this in the manuscript.

- Please determine the anomeric ratio (α:β) of the products by NMR analysis.

- More evidence on the proposed mechanism shown in Fig. 8 is required, as no antecedent appears in the cited reference.

Author Response

The comments from Reviewer #2:

In this article, Te-Sheng Chang and co-workers report the glycosylation of triterpenoid aglycones via DgAS catalysis. All compound structures were supported by spectroscopic data and the manuscript is easy to follow although some spelling errors were found. Having said the above, I recommend the manuscript for publication in Biology after addressing a number of recommendations:

  1. “It would be desirable to include in the introduction some comparative examples of saponins that display greater biological activity than their aglycones.”

Response:

“In plants, triterpenoids usually exist in the glycosidic forms, called saponins, with linkages of sugar moieties. Some saponins possess unique bioactivities while their aglycone forms do not. For example, ginseng saponin Rg1 has been reported to increase human immune responses, detox oxidized low-density-lipoproteins, relax the endothelium-dependent porcine coronary arteries, improve insulin resistance in diabetes, suppress tumor cell growth in cancers and arrest melanoma cell cycle progression in the G1 phase. Animal model studies have also demonstrated that ginseng saponins Rb1, CK and Rg1 can prevent scopolamine-induced memory deficits. In contrast, the ginseng triterpenoid aglycones did not possess the bioactivities [reviewed in 6].”

In order to explain clearer, we added the above description at lines from 65 to 73 at page 2 in the Introduction of the revised manuscript.

  1. “To which structural characteristics in the substrate is attributed that the reaction does not proceed with the remaining triterpenoids? was only the standardized method evaluated or were different conditions also tested?”

Response:

“According to the proposed mechanism, the lack of free carboxyl group in GAA-26-O-b-glucoside explains why the molecule cannot be glycosylated by DgAS (Table 1). Also, an oxo group (C-23) near the glycosylated carboxyl group (C-26) is needed for the stabilization of the intermediate (Figure 8). Thus, neither antcin K nor celastrol is glycosylated by DgAS due to the lack of the oxo group near their carboxyl group. In addition, the bulky glucosyl group at C-15 of GAA-15-O-b-glucoside might form steric hindrance in the catalytic reaction by DgAS and block the catalytic activity toward the molecule by DgAS.”

In order to explain clearer, we added the above description at lines from 335 to 342 at page 15 in the Results and Discussion sec. 3.5 of the revised manuscript.

  1. Please, detail the criteria for the selection of the 7 aglycones, particularly antcin K and

Response:

“Based on the novel glycosylation activity by DgAS toward the carboxyl group of the tetracyclic triterpenoid GAF, other either tetracyclic or pentacyclic triterpenoids (antcin K, celastrol and 4 ganoderic acids in Figure 4) with carboxyl group were selected to evaluate detailed acidic glycosylation activity of DgAS.”

In order to explain clearer, we added the above description at lines from 235 to 238 at page 9 in the sec. 3.3 of the revised manuscript.

  1. “Were studies conducted to determine the number of catalytic cycles given by the enzyme before its inactivation? Please indicate this in the manuscript.”

Response:

We did not evaluate the number of catalytic cycles by the enzyme before its inactivation. We added the indication at lines from 189 to 190 at page 6 in the sec. 3.1 of the revised manuscript.

  1. “Please determine the anomeric ratio (α:β) of the products by NMR analysis.”

Response:

Thank you for the comments. “According to the NMR analysis, the anomeric ratio (a:b) of the products was 3:2 (60%:40%).” We added the description at lines from 228 to 229 at page 8 in the sec. 3.2 and from 320 to 321 at page 14 in the sec. 3.4 of the revised manuscript.

  1. “More evidence on the proposed mechanism shown in Fig. 8 is required, as no antecedent appears in the cited reference.”

Response:

Thank you very much for the comment. The new mechanism of glycosylation by amylosucrose in the present is indeed worth to be studied in advance, especially by structural analysis. We are searching for the cooperators to do.

Reviewer 3 Report

The paper by Wu et al reports a new mechanism of glycosylation by amylosucrose to produce glycoside monomers. I find the study to be of interest to the readers of biology. The study is well done and uses sound scientific design to investigate the glycosylation.  I recommend publication.

Author Response

The comments from Reviewer #3:

The paper by Wu et al reports a new mechanism of glycosylation by amylosucrose to produce glycoside monomers. I find the study to be of interest to the readers of biology. The study is well done and uses sound scientific design to investigate the glycosylation.  I recommend publication.

Response:

Thank you very much for the comment.

Reviewer 4 Report

 This study proposed a new mechanism to form the glycoside anomers of triterpenoids such as Ganoderma triterpenoid saponin anomers by DgAS,which could be applied for new triterpenoid derivatives in pharmaceutical biotechnology. The conclusions are supported by the experiments. However, the conclusion should be further confirmed by structral information or at least by modelling. How could the DgAs contact and convert the substrate?

Minor concerns:

1.The fonts of the coordinate axes in figure 3 need to be increased.

2.The resolution of Figures 1, 4 and 5 is low.

3. Please combine the figures into 5 or 6 figures. 

Author Response

The comments from Reviewer #4:

 “This study proposed a new mechanism to form the glycoside anomers of triterpenoids such as Ganoderma triterpenoid saponin anomers by DgAS,which could be applied for new triterpenoid derivatives in pharmaceutical biotechnology. The conclusions are supported by the experiments. However, the conclusion should be further confirmed by structural information or at least by modelling. How could the DgAs contact and convert the substrate?”

Response:

Thank you very much for the comment. The new mechanism of glycosylation by amylosucrose in the present is indeed worth to study. We are still working on it due to some technical problems.

  1. “The fonts of the coordinate axes in figure 3 need to be increased.”

Response:

Thank you very much for the comment. We revised and enlarged the coordinate axes in figure 2 (the original figure 3) in the revised manuscript.

  1. “The resolution of Figures 1, 4 and 5 is low.”

Response:

We increase the resolution of figure 1, 3, and 4 (the original 1, 4, and 5) in the revised manuscript.

  1. Please combine the figures into 5 or 6 figures. 

Response:

We moved the figure 2 and figure 6 of the original manuscript to the supplemental materials as figure S1 and figure S10 in the revised manuscript. In the revised version, there are 6 figures in the main text.  
